# Blue Light Exposure Caused Large-Scale Transcriptional Changes in the Abdomen and Reduced the Reproductive Fitness of the Fall Armyworm *Spodoptera frugiperda*

**DOI:** 10.3390/insects15010010

**Published:** 2023-12-26

**Authors:** Yu Liu, Yi-Dong Tao, Li-Bao Zhang, Fen Wang, Jin Xu, Jun-Zhong Zhang, Da-Ying Fu

**Affiliations:** 1Key Laboratory of Forest Disaster Warning and Control in Yunnan Province, Faculty of Biodiversity Conservation, Southwest Forestry University, Kunming 650224, China; liuyu32098@163.com (Y.L.); yidongtao2022@163.com (Y.-D.T.); wangfen0208@163.com (F.W.);; 2Yunnan Key Laboratory of Plateau Wetland Conservation, Restoration and Ecological Services, Southwest Forestry University, Kunming 650224, China; 3Tianbao Customs Comprehensive Technical Center, Wenshan 663603, China

**Keywords:** *Spodoptera frugiperda*, blue light stress, survival, reproduction, stress detection and response, oxidative stress

## Abstract

**Simple Summary:**

The study of blue light stress and the response mechanism in insects helps in the prediction of the harm to human health from blue light and in the control of pests. Here, we found that blue light exposure negatively affected survival and reproduction in *Spodoptera frugiperda*, with a longer exposure resulting in stronger effects. Further study in female adults found that blue light exposure caused remarkable transcriptional changes in the head, thorax and abdomen. A functional analysis revealed that most stress detection-, response- and defense-related genes or pathways were upregulated in the head and thorax with a shorter duration of blue light exposure. In the abdomen, blue light exposure resulted in the downregulation of a large number of genes, including many egg-development-related genes, which may be partially related to the reduced fecundity in blue-light-stressed females. This is the first study to test blue light stress-induced transcriptional changes in the thorax and abdomen of insects, providing basic data for understanding the molecular response mechanisms of moths under blue light stress.

**Abstract:**

In the present study, we found that blue light stress negatively affected the development periods, body weight, survival and reproduction of *Spodoptera frugiperda*, and it showed a dose-dependent reaction, as longer irradiation caused severer effects. Further transcriptome analysis found blue light stress induced fast and large-scale transcriptional changes in the head, thorax and, particularly, the abdomen of female *S. frugiperda* adults. A functional enrichment analysis indicated that shorter durations of blue light irradiation induced the upregulation of more stress response- and defense-related genes or pathways, such as abiotic stimuli detection and response, oxidative stress, ion channels and protein-kinase-based signal pathways. In the abdomen, however, different durations of blue-light-exposure treatments all induced the downregulation of a large number genes and pathways related to cellular processes, metabolism, catalysis and reproduction, which may be a trade-off between antistress defense and other processes or a strategy to escape stressful conditions. These results indicate irradiation duration- and tissue-specific blue light stress responses and consequences, as well as suggest that the stress that results in transcriptional alterations is associated with the stress that causes a fitness reduction in *S. frugiperda* females.

## 1. Introduction

A large number of studies have documented the ecological consequences of artificial light on animal physiology, psychology and behavior, such as preventing some nocturnal animals from foraging, causing habitat fragmentation and aggregating insects making them vulnerable to predation, as well as reducing the population of insects [1,2,3,4,5]. Studies also show that some short-wavelength lights, such as ultraviolet (UV) light and blue light, have lethal effects on insects [6,7,8]. Thus, in agriculture, artificial lights have been widely used for pest monitoring and management [9].

Blue light is short-wavelength (400–500 nm) visible light and it has more energy than green and red light [10]. Nowadays, blue light is ubiquitous in our everyday lives, as it can be produced by computer monitors, mobile phones, fluorescent and LED lights, digital displays and products, etc. Because of the increase in the amount of work conducted at nighttime and the widespread application of digital displays and LED lights, the frequency and duration of exposure to blue light are continuously increasing [11]. The negative impacts on human health caused by blue light stress have, thus, obtained widespread concern in recent years [12]. Although the effects of long-range blue light exposure on humans are still to be clarified, the toxicity of blue light has been confirmed in cells and organisms such as mice, nematodes and fruit flies [13,14,15]. Hori et al. [16] found that blue-light stress can cause death in mosquitoes, fruit flies and flour beetles and they found that these effective wavelengths were species-specific. Later studies in the strawberry leaf beetle showed that the irradiation of eggs with 438 nm (peak wavelength) light caused 90% of the eggs to die before hatching. In addition, up to 55% of pupae treated with 407–465 nm lights died before eclosion [6]. In *D. melanogaster*, the 466 nm (peak wavelength) light was the most harmful to pupae and the 417 nm light had the strongest lethal effect on adults under low light intensities, but 466 nm had the greatest effect on adults under high light intensities [17]. Molecular studies showed that blue-light-stress-induced cells, including retinal cells and nonretinal cells, work to synthesize reactive oxygen species (ROS) and may also induce apoptosis in cells [18,19]. A study in porcine retinal pigment epithelium cells showed that blue light stress increased levels of ROS and decreased superoxide dismutase (SOD) and catalase activities, as well as decreased the mitochondrial membrane potential and respiratory activities [20]. Studies in *Drosophila* and mice showed that a single critical blue light irradiation resulted in the death of photoreceptors in the retina [21,22]. In *C. elegans*, visible light, particularly blue light, can incur oxidative stress and even reduce longevity [23]. Compared to flies raised in the absence of blue light or in continuous darkness, fruit flies raised in the 12 h cycle of blue light and darkness exhibited neurodegeneration in the brain and shortened lifespans [24]. And a recent study showed that these effects are age-dependent, with older flies being more sensitive to blue light [25].

The head of an insect houses sensory organs and the brain, which play a key role in decision making and taking action, as well as regulating complex biological processes and behaviors [26]. Insects usually have two forms of photoreceptive organs: ocelli and compound eyes [27]. A compound eye usually has three categories of photoreceptor cells, which have a wide spectral sensitivity, including the blue wavelength region [28]. The insect brain consists of three pairs of lobes, which are fused ganglia and clusters of neurons [29]. A great number of neuropeptides and their receptors have been found in the brain of insects [30]. These neuropeptides have multiple functions, such as energy metabolism, osmoregulation and behavior. For example, a neuropeptide known as natalisin was identified in the brain’s neurons, which regulates mating behavior and fecundity in *D. melanogaster* and *Tribolium castaneum* [31]; *Bactrocera dorsalis* [32]; and *Spodoptera litura* [33]. The above studies imply that blue light irradiation may have a directly toxic impact on photoreceptors [21,22] and the brain [24] in insects, which then leads to further negative effects on the organismal level. However, further studies are required to elucidate this hypothesis and the related mechanisms.

The above studies also suggest that blue light may cause damage to parts of the body other than the head, such as the thorax and abdomen. The insect thorax is almost exclusively adapted for locomotion; it contains three pairs of legs and, in many adult insects, one or two pairs of wings. Flight ability is essential for the survival and reproduction of insects, which have evolved relatively faster throughout the class [34,35]. The abdomen is the reproductive center of insects, with the ovaries occupying the majority of its space. The abdomen also contains a large amount of fat bodies [36], which are the main source of energy and nutrients required for survival and reproduction. However, empirical studies are required to determine how blue light causes damage to the thorax and abdomen. Recent studies on reproduction- and stress-induced (such as UV, thermal and chemical stress) responses in different tissues of different insect species based on RNA-seq and bioassay [37,38,39,40] have provided a rich resource to test the blue light stress response and possible mechanisms in different tissues of insects.

The progress of high-throughput sequencing techniques and bioinformatics in recent years has provided a broader perspective and deeper insights into insect molecular processes and mechanisms. A recent study in the Asian citrus psyllid *Diaphorina citri* by using transcriptome analysis has suggested that blue light stress caused physiological and molecular responses, such as inflammation, protein denaturation, oxidative stress and tumor growth [37].

Taking all aforementioned factors into account, conducting research on the response and consequence of insects to blue light irradiation will not only help us to understand blue-light-stress-related mechanisms, but can also provide ideas and theoretical support for the control of insect pests. A recent study has shown that blue and UV light can be used to distract the greenhouse whitefly (*Trialeurodes vaporariorum*) from its host [41].

The fall armyworm, *S. frugiperda*, is native to tropical and subtropical regions of the Americas. Although *S. frugiperda* is a long-distance migratory pest, there were no reports of its distribution outside the Americas before 2015. *S. frugiperda* was first detected in Africa in 2016 [42] and then in India and in southern Asia in 2018 [43,44], followed by southwesten China at the end of 2018 and then it quickly spread to a large area of China [45]. *S. frugiperda* is now invading Oceania [46]. This pest prefers to feed on corn and can cause huge economic losses [47]. This moth pest is also featured for its long-distance migration capacity [48], high reproductive potential [49] and strong drug resistance [50,51]. Sustainable and environmentally friendly control strategies are imperative for better control of this moth pest.

Therefore, in the present study, we hypothesized that blue light exposure would have negative effects on the fitness of moths and generate a transcriptional response not only in the head but also in other body parts of adults. To test this hypothesis, we set up blue light treatments in *S. frugiperda* to test how blue light exposure affects development and reproductive fitness, as well as the transcriptional changes in the head, thorax and abdomen of adult females. This study also contributes to the development of environmentally friendly control strategies for *S. frugiperda*.

## 2. Materials and Methods

### 2.1. Insects

*S. frugiperda* larvae were collected on corn plants in the farmland near the Dongchuan county of Yunnan Province, China. Collected larvae were reared on an artificial diet [40] under 27 ± 1 °C, 60–80% relative humidity and a photoperiod of 14:10 h (light/dark). To ensure virginity, mature pupae were sexed based on morphological characteristics [52] and then female and male pupae were separately caged for eclosion. Newly eclosed female and male adults were collected and separately reared under the above conditions and fed with a 10% honey solution. Under the above rearing condition, the life cycle of *S. frugiperda* is about 3 d for eggs, 15 d for larvae (have 6 instars), 8 d for pupae and 10 d for adults, respectively.

### 2.2. Effect of Blue Light Exposure on the Development and Reproduction of S. frugiperda

Three blue light treatments (T1, T2 and T3; based on exposure duration) and one control were performed to test the effect of blue light exposure on the development and reproductive fitness of *S. frugiperda*: T1—insects were reared under normal light condition (14 h normal white light/10 h dark) from egg to pupal stage, while adults were irradiated by blue light for 14 h each day (14 h blue light/10 h dark) from eclosion to 3 d old (short-duration irradiation); T2—insects were irradiated by blue light for 7 h each day (7 h blue light/7 h normal white light/10 h dark) from neonatal egg to 3 d old adult (moderate-duration irradiation); T3—insects were irradiated by blue light for 14 h each day (14 h blue light/10 h dark) since neonatal egg to 3 d old adult (long-duration irradiation); control—insects were reared under normal light condition (14 h normal white light/10 h dark) from egg to adult. The blue light was provided by a 30 W blue light LED bulb (Jiadeng Lighting, China; peak wavelength, 465.4 nm, spectral bandwidth, 18.6 nm, measured by a spectrometer, HPCS330P, Hopoocolor, Hangzhou, China) and normal light by a 5 W white light LED bulb (Jiadeng Lighting, Hangzhou, China; broad spectrum, 400–700 nm). The light exposure treatments were conducted in a light cage (40 × 40 × 40 cm) by setting the bulb on the top of the cage according to previous studies [37,53]. The reason for using different wattage bulbs was to ensure comparable light intensity. The light intensities in all these experiments were about 190–200 lux and were measured using a portable luxmeter (Smart Sensor, Shanghai, China). Insects were reared in transparent plastic boxes (see below) and the light intensities were measued in the boxes under its transparent plastic cover.

Following the above treatments, the duration of larval and pupal stages, the bodyweight of mature larvae (wandering larvae) and newly emerged pupae and adults, pupation rate (percent of number of pupae/number of start larvae), eclosion rate (percent of number of adults/number of pupae) and survival rate from larvae to adults (percent of number of adults/number of start larvae) were recorded. To test the pupation rate, eclosion rate and survival rate, three replicates were used for each treatment, with 120 larvae per replicate. Larvae were reared in transparent plastic boxes (20 cm long, 14 cm wide and 4 cm high) with cells (each box has 24 cells; the size of each cell is 4.5 cm long, 2.4 cm wide and 3.8 cm high), with one cell one insect to avoid cannibalism and facilitate observations. Some of these insects were also used (via random selection) for the following tests. To test the developmental duration, one insect was used as a replicate and 30 larvae and 60 (30 male and 30 female) pupae were randomly selected from each treatment for the measurements. To test the bodyweight of different stages, one insect was used as a replicate, and 30 larvae, 60 (30 male and 30 female) pupae and 60 (30 male and 30 female) adults were randomly selected from each treatment for the measurements.

Three-day-old virgin male and female adults from each treatment were collected and paired in plastic boxes (25 cm long, 15 cm wide and 8 cm high; one pair per box) for mating and oviposition under normal conditions (without blue light exposure). Each box had a paper strip (15 × 20 cm) folded in zigzag pattern as the oviposition substrate and a 10% honey solution as food. Eggs were collected and incubated in petri dishes (8.5 × 1.5 cm) under normal conditions. The number of hatched eggs (offspring larvae) was recorded 4 days after incubation. Twenty pairs were used for each treatment (n = 20).

Significant differences between treatments regarding the data of development and reproduction were analyzed using an ANOVA followed by Tukey’s studentized range test for multiple comparisons. Prior to analysis, data were tested for normality using the Shapiro–Wilk test and for homogeneity of variances using Levene’s test across the treatments. Percentage data were arcsine square root-transformed to normalize the data before the test. The rejection level was set at *α* < 0.05. The values reported here are means ± SE.

### 2.3. Blue Light Exposure Induced Transcriptional Changes in S. frugiperda

#### 2.3.1. Treatments and Sampling

Three-day-old virgin female moths were collected following the above treatments and their heads, thoraxes and abdomens were sampled separately for RNA sequencing. Ten heads, ten thoraxes or ten abdomens were combined to form a head, thorax or abdomen sample replicate of each treatment and three replicates were used for each sample. All samples were frozen with liquid nitrogen immediately after collection and then were stored at −80 °C.

#### 2.3.2. cDNA Library Preparation and Sequencing

Total RNAs of samples were extracted using the TRIzol reagent (Invitrogen, Waltham, MA, USA) according to the manufacturer’s instructions. RNA concentration and purity were measured using a spectrophotometer (IMPLEN, Westlake Village, CA, USA) and a Qubit RNA Assay Kit (Life Technologies, Carlsbad, CA, USA). RNA integrity was assayed using an RNA Nano 6000 Assay Kit (Agilent Technologies, Santa Clara, CA, USA). Sequencing libraries were developed using the NEBnext Ultra RNA Library Prep Kit for Illumina (New England BioLabs, Ipswich, MA, USA) and sequences of each sample were labeled by adding index codes. The prepared libraries were sequenced using an Illumina HiSeq 4000 platform (Illumina, Foster City, CA, USA) to obtain 125–150 bp paired-end reads.

The raw reads from the above sequencing were screened to obtain clean reads by deleting low-quality reads and the adapter in the reads. The Q20 and Q30 values of the clean data were also calculated. The clean reads were then mapped to the reference genome sequence of *S. frugiperda* [54] by using the Hisat2 software (v2.1.0).

#### 2.3.3. Differential Expression Analysis and Functional Annotation

Transcript levels were measured using the TPM (transcripts per million) method. The differential expression between treatments was analyzed using DESeq2 (v1.24.0). The *p* value was adjusted to the *q*-value [55] and |log2(foldchange)| > 1 and *q* < 0.05 were used as the thresholds for determining significantly different expression.

Differentially expressed genes (DEGs) were compared with databases using BLAST software (v2.9.0) for functional annotation [56]. GOSeq (v2.12) and KOBAS (v2.1.1) were used for GO and KEGG enrichment analysis, respectively. And *q* < 0.05 was set as the threshold to determine significantly enriched GO terms or KEGG pathways.

#### 2.3.4. Validation of RNAseq Results via qPCR

Twenty-nine DEGs were selected for qPCR analysis and the *GAPDH* (ID: LOC118271716) was used as a reference gene. Total RNAs of samples were isolated using RNAiso plus (TaKaRa, Beijing, China) and cDNA for qPCR was prepared using PrimeScript RT kit (TaKaRa, Beijing, China). PCR was conducted with the QuantStudio 7 Flex System (Thermo Fisher Scientific, Waltham, MA, USA) using gene-specific primers (Appendix A) and a program of 95 °C for 30 s, and then 40 cycles of 95 °C for 5 s, 60 °C for 30 s and dissociation. The relative expression was calculated using the 2^−ΔΔCT^ method [57]. Differences in gene expression between samples were analyzed using ANOVA as above. All values were reported as mean ± SE. 

## 3. Results

### 3.1. Effect of Blue Light Exposure on the Developmental Periods and Survival of S. frugiperda

Blue light exposure significantly affected the larval developmental period (*F*_3,116_ = 4.59, *p* < 0.0001; Figure 1a). Post hoc analysis showed that T2 and T3 had a significantly (*p* < 0.05) longer larval period than T1 and the control. However, no significant difference on the female pupal period (*F*_3,116_ = 2.04, *p* = 0.113; Figure 1b) or male pupal period (*F*_3,116_ = 0.26, *p* = 0.856; Figure 1c) was found between treatments.

Blue light exposure also significantly affected the pupation rate (*F*_3,8_ = 25.11, *p* < 0.0001; Figure 1d), eclosion rate (*F*_3,8_ = 23.57, *p* < 0.0001; Figure 1e) and survival rate (*F*_3,8_ = 27.34, *p* < 0.0164; Figure 1f). Post hoc analysis showed that T2 and T3 have significantly (*p* < 0.05) lower pupation rates and survival rates than those of T1 and the control. T1 showed the highest eclosion rate, followed by the control and T2, and T3 showed the lowest eclosion rate.

### 3.2. Effect of Blue Light Exposure on the Bodyweight and Reproduction of S. frugiperda

Moreover, blue light exposure significantly affected larval (*F*_3,116_ = 4.59, *p* = 0.005), pupal (*F*_3,116_ = 14.79, *p* < 0.0001 for female; *F*_3,116_ = 17.62, *p* < 0.0001 for male) and adult (*F*_3,116_ = 23.6, *p* < 0.0001 for female; *F*_3,116_ = 11.8, *p* < 0.0001 for male) bodyweight (Figure 2a–e). The post hoc analysis showed that T3 has significantly lower larval bodyweight than that of the control and T1 (*p* < 0.05) and T2 also showed a lower larval bodyweight than the control and T1, but this was not significant (*p* > 0.05). In male and female pupae and female adults, T2 and T3 showed significantly lower bodyweights than those of the control and T1 (*p* < 0.05) and T3 showed a lower bodyweight than T2, but this was not significant (*p* > 0.05). Control adult males had the highest bodyweight, followed by those of T1 and T2 and T3 had the lowest bodyweight (*p* < 0.05).

Blue light exposure showed significant effects on the number of eggs laid (*F*_3,76_ = 3.45, *p* = 0.021), number of hatched eggs (*F*_3,76_ = 7.45, *p* < 0.0001) and egg hatching rate (*F*_3,76_ = 6.24, *p* = 0.001) in *S. frugiperda* females (Figure 2f–h). Post hoc analysis showed that all treatments have more or less the same effects on the number of eggs laid, number of hatched eggs and egg hatching rate, with T3 females showing the lowest values for these parameters (*p* < 0.05).

### 3.3. Blue-Light-Exposure-Induced Transcriptional Changes

#### 3.3.1. Sequencing Quality

RNAseq achieved up to 98,300,000 clean reads from each of the libraries (Appendix A). Further calculation showed that the Q20 values of each the libraries ranged from 96.24% to 98.04%, while the Q30 ranged from 90.40% to 94.14%. Clean reads were mapped to the genome of *S. frugiperda* and the mapped ratios of each of the libraries ranged from 77.02% to −83.79%. Pearson’s correlation coefficient (Appendix A) and PCA (principal component analysis) (Appendix A) verified the reproducibility of biological replicates and the sequencing. The raw reads from RNAseq were uploaded onto the SRA database of NCBI (accession no.: PRJNA1027141).

#### 3.3.2. Summary of Differential Expression Analysis

Blue light exposure induced remarkable transcription changes in the *S. frugiperda* female head, which showed 163, 488 and 58 DEGs in T1, T2 and T3 female head (T1H, T2H and T3H) compared to that of the control female head (CKH), respectively (Figure 3a–c; Appendix A). Blue light exposure induced 284, 145 and 903 DEGs in T1, T2 and T3 female thorax (T1T, T2T and T3T) compared to that of the control female thorax (CKT), respectively (Figure 3d–f; Appendix A). Blue light exposure induced much more transcriptional changes in the female abdomen, which showed 3358, 1784 and 2642 DEGs in the T1, T2 and T3 female abdomen (T1A, T2A and T3A) compared to that of the control female abdomen (CKA), respectively (Figure 3g–i; Appendix A).

There are 12, 38 and 1406 common DEGs among the three treatment comparison groups of the head, thorax and abdomen, respectively (Appendix A), and 6, 7 and 7 common DEGs among the three tissue comparison groups of the T1, T2 and T3 treatments, respectively (Appendix A).

These DEGs were significantly (*q* < 0.05) enriched to 844 GO terms and 167 KEGG pathways (Appendix A), with most (532/844 = 63.03%) terms belonging to biological process (BP) and many (53/167 = 31.74%) pathways belonging to human diseases. For summary and easy understanding, significantly enriched terms/pathways were divided into 13 classes according to their function (Figure 4, Figure 5 and Figure 6). To provide more related information, the top 20 enriched terms or pathways of each of the comparison groups were also presented (Appendix A) and discussed for those comparison groups that had few or many more terms/pathways. For those groups that did not have 20 significantly enriched terms or pathways, insignificantly (*q* > 0.05) enriched terms or pathways were also included. Based on these analyses, the important tissue-specific blue-light-stress-induced transcriptional changes were investigated and discussed in detail as seen below.

#### 3.3.3. Blue-Light-Exposure-Induced Transcriptional Changes in Female Head

Blue light exposure induced more upregulated DEGs (131, 309 and 44 for T1, T2 and T3 treatments, respectively) and less downregulated DEGs (32, 179 and 14 for T1, T2 and T3 treatments, respectively) in female heads (Figure 3a–c). 

T1 and T3 upregulated DEGs (up-DEGs) did not significantly enrich any terms/pathways. Within the top 20 terms/pathways enriched in T1 up-DEGs (*q* > 0.05), 3 relating to stress response and defense (e.g., bile secretion), 2 to reproduction (e.g., oxytocin signaling pathway), 2 to immunity (e.g., melanogenesis) and 7 to disease (e.g., Kaposi sarcoma-associated herpesvirus infection) terms/pathways were found (Appendix A). Within the top 20 terms/pathways enriched in T3 up-DEGs (*q* > 0.05), 12 relating to disease (e.g., small cell lung cancer), 7 to immunity and 8 to catalysis (e.g., peroxisome organization) terms/pathways were found (Appendix A). T2 up-DEGs were significantly (*q* < 0.05) enriched to 50 terms/pathways, with 1 relating to detecting stress (thermogenesis), 1 to stress response and defense (oxidative phosphorylation), 19 to metabolism (e.g., cellular nitrogen compound metabolic process and selenocompound metabolism), 15 to cellular process (e.g., mitochondrial protein complex and cardiac muscle contraction), 8 to disease (e.g., Parkinson’s disease), etc. (Figure 4b, Appendix A).

**Figure 4 insects-15-00010-f004:**
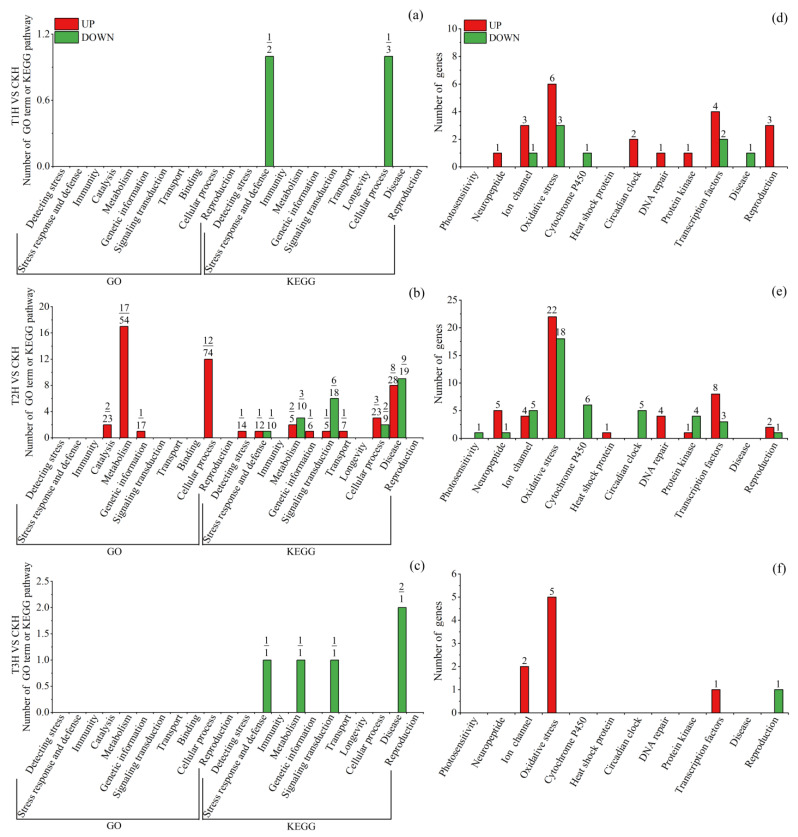
Significantly (*q* < 0.05) enriched GO terms and KEGG pathways, and DEGs directly related to blue light exposure response and reproduction in *S. frugiperda* female heads. (**a**–**c**) are terms/pathways of T1, T2 and T3 treatments, respectively. Red columns denote terms or pathways enriched in upregulated DEGs, while green columns denote terms or pathways enriched in downregulated DEGs. At the top of each column, the upper number is the number of terms or pathways, while the lower number is the number of enriched DEGs. (**d**–**f**) are DEGs of T1, T2 and T3 treatments, respectively. Red columns denote upregulated DEGs, while green columns denote downregulated DEGs. The number at the top of each column represents the number of DEGs. T1H, T2H and T3H refer to female head from T1, T2 and T3 treatments, respectively; T1T, T2T and T3T refer to female thorax from T1, T2 and T3 treatments, respectively; T1A, T2A and T3A refer to female abdomen from T1, T2 and T3 treatments, respectively; CKH, CKT and CKA refer to control female head, thorax and abdomen, respectively; VS, versus.

T1 and T3 downregulated DEGs (down-DEGs) were only significantly enriched to two and five pathways, respectively (Figure 4a,c). In the top 20 terms/pathways enriched in T1 down-DEGs (including significant and insignificant terms/pathways), 3 relating to stress response and defense (such as oxidative phosphorylation), 5 to catalysis (such as regulation of phosphoprote in phosphatase activity) and 5 to disease (such as Parkinson’s disease) terms/pathways were found (Appendix A). In the top 20 terms/pathways enriched in T3 down-DEGs (including significant and insignificant terms/pathways), about half relating to metabolism, 4 to disease (such as amphetamine addiction) and 2 terms/pathways to reproduction (folate biosynthesis and prolactin signaling pathway) were found (Appendix A). T2 down-DEGs significantly enriched 21 pathways, with 3 relating to metabolism (such as fructose and mannose metabolism), 6 to the signaling pathway (such as Hippo signaling pathway—fly; cGMP-PKG signaling pathway) and 9 to disease (such as Prion disease), etc. (Figure 4b, Appendix A).

Specific DEGs that may directly relate to blue light exposure response and reproduction were also identified and presented (Figure 4d–f, Appendix A). Relatively more oxidative-stress-related DEGs were found, with 9 (6 up, 3 down), 40 (22 up, 18 down) and 5 (up) DEGs for T1, T2 and T3, respectively (Figure 4d–f). A number of neuropeptide-, ion channel-, DNA repair- and transcription factor-related DEGs were also found, with most of them being upregulated.

#### 3.3.4. Blue-Light-Exposure-Induced Transcriptional Changes in Female Thorax

Blue light exposure also induced more up-DEGs (200, 87 and 551 for T1, T2 and T3 treatments, respectively) and less down-DEGs (84, 58 and 352 for T1, T2 and T3 treatments, respectively) in female thoraxes (Figure 3d–f).

T1 up-DEGs were significantly enriched to 31 terms/pathways, with 4 relating to detecting stress (e.g., detection of light stimulus and detection of external stimulus), 7 to stress response and defense (e.g., response to radiation and response to light stimulus), 6 to signal transduction (e.g., intracellular signal transduction; phototransduction—fly), etc. (Figure 5a, Appendix A). T2 up-DEGs were significantly enriched to nine pathways, with one relating to immunity, one to longevity (longevity regulating pathway—multiple species), three to disease (e.g., toxoplasmosis), etc. (Figure 5b, Appendix A). T3 up-DEGs were significantly enriched to seven terms/pathways, with three relating to catalysis (e.g., transferase activity and transferring acyl groups), two to binding (e.g., phosphopantetheine binding), etc. (Figure 5c, Appendix A).

**Figure 5 insects-15-00010-f005:**
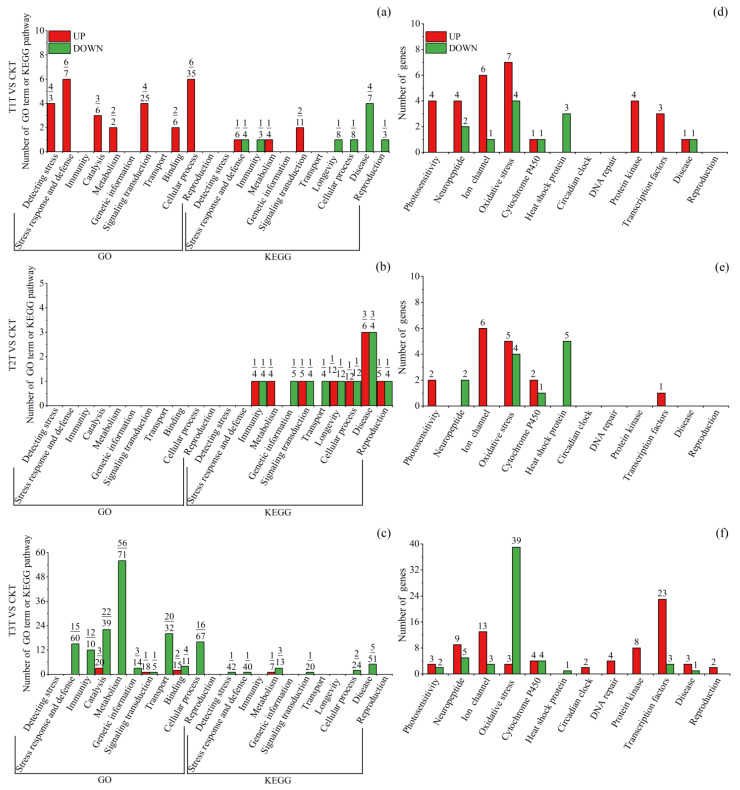
Significantly (*q* < 0.05) enriched GO terms and KEGG pathways, and DEGs directly related to blue light exposure response and reproduction in *S. frugiperda* female thoraxes. (**a**–**c**) are terms/pathways of T1, T2 and T3 treatments, respectively. Red columns denote terms or pathways enriched in upregulated DEGs, while green columns denote terms or pathways enriched in downregulated DEGs. At the top of each column, the upper number is the number of terms or pathways, while the lower number is the number of enriched DEGs. (**d**–**f**) are DEGs of T1, T2 and T3 treatments, respectively. Red columns denote upregulated DEGs, while green columns denote downregulated DEGs. The number at the top of each column represents the number of DEGs. T1H, T2H and T3H refer to female head from T1, T2 and T3 treatments, respectively; T1T, T2T and T3T refer to female thorax from T1, T2 and T3 treatments, respectively; T1A, T2A and T3A refer to female abdomen from T1, T2 and T3 treatments, respectively; CKH, CKT and CKA refer to control female head, thorax and abdomen, respectively; VS, versus.

T1 down-DEGs were significantly enriched to nine terms/pathways, with one relating to stress response and defense, one to immunity, one to longevity (longevity regulating pathway—multiple species), four to disease (such as toxoplasmosis), etc. (Figure 5a, Appendix A). T2 down-DEGs were significantly enriched to ten terms/pathways, with one relating to genetic information, one to signaling transduction, one to longevity (longevity regulating pathway—multiple species) and three to disease (such as Legionellosis), etc. (Figure 5b, Appendix A). T3 down-DEGs were significantly enriched to 162 terms/pathways, with 16 relating to stress response and defense (such as response to external stimulus and defense response), 12 to immunity (such as antibacterial humoral response and immune response), 22 to catalysis (such as threonine-type peptidase activity), 59 to metabolism (such as pyruvate metabolism and purine ribonucleoside triphosphate metabolic process), 20 to transport (such as transmembrane transport), 18 to cellular process (such as cardiac muscle contraction), etc. (Figure 5c, Appendix A). In the top 20 terms/pathways enriched in T3 down-DEGs, about half were related to metabolism and one longevity-related (longevity regulating pathway—multiple species) pathway was found (Appendix A). 

On specific gene level, more oxidative-stress-related DEGs were found, with 11 (7 up, 4 down), 9 (5 up, 4 down) and 42 (3 up, 39 down) DEGs for T1, T2 and T3, respectively (Figure 5d–f). DEGs related to photosensitivity, neuropeptide, ion channels, P450, DNA repair, protein kinase and transcription factors were also found, with most of them being upregulated. A total of nine heat shock protein (HSP) DEGs were found but all were downregulated.

#### 3.3.5. Blue-Light-Exposure-Induced Transcriptional Changes in Female Abdomen

Blue light exposure induced less up-DEGs (1208, 450 and 887 for T1, T2 and T3 treatments, respectively) but more down-DEGs (2150, 1334 and 1755 for T1, T2 and T3 treatments, respectively) in female abdomens (Figure 3g–i), which is contrary to the results of the head and thorax. 

T1 up-DEGs were significantly enriched to 45 terms/pathways, with 4 relating to stress response and defense (e.g., cellular response to DNA damage stimulus), 5 to catalysis (e.g., methyltransferase activity), 10 to metabolism (e.g., organic cyclic compound metabolic process), 5 to binding (e.g., nucleic acid binding), etc. (Figure 6a, Appendix A). T2 up-DEGs were significantly enriched to 18 terms, with 2 relating to catalysis (e.g., catalytic activity and acting on RNA), 11 to metabolism (e.g., cellular macromolecule metabolic process), etc. (Figure 6b, Appendix A). T3 up-DEGs were significantly enriched to 47 terms/pathways, with 8 relating to catalysis (e.g., transferase activity and transferring one-carbon groups), 13 to metabolism (e.g., organic cyclic compound metabolic process), 6 to genetic information, etc. (Figure 6c, Appendix A).

**Figure 6 insects-15-00010-f006:**
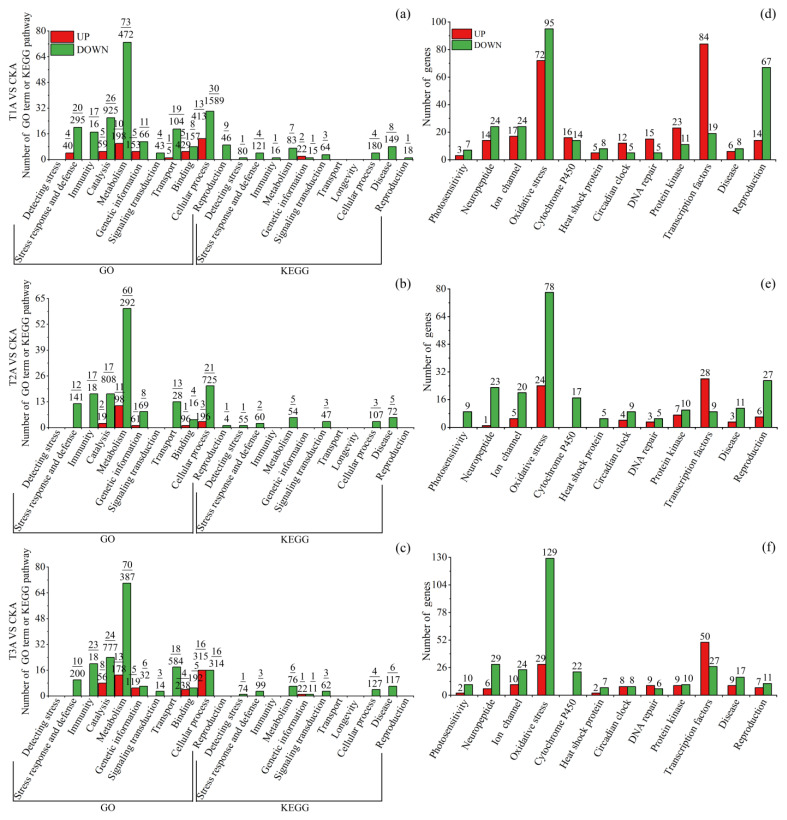
Significantly (*q* < 0.05) enriched GO terms and KEGG pathways, and DEGs directly related to blue light exposure response and reproduction in *S. frugiperda* female abdomens. (**a**–**c**) are terms/pathways of T1, T2 and T3 treatments, respectively. Red columns denote terms or pathways enriched in upregulated DEGs, while green columns denote terms or pathways enriched in downregulated DEGs. At the top of each column, the upper number denotes the number of terms or pathways, while the lower number denotes the number of enriched DEGs. (**d**–**f**) are DEGs of T1, T2 and T3 treatments, respectively. Red columns denote upregulated DEGs, while green columns denote downregulated DEGs. The number at the top of each column denotes the number of DEGs. T1H, T2H and T3H refer to female head from T1, T2 and T3 treatments, respectively; T1T, T2T and T3T refer to female thorax from T1, T2 and T3 treatments, respectively; T1A, T2A and T3A refer to female abdomen from T1, T2 and T3 treatments, respectively; CKH, CKT and CKA refer to control female head, thorax and abdomen, respectively; VS, versus.

T1 down-DEGs were significantly enriched to 247 terms/pathways, with 24 relating to stress response and defense, 18 to immunity, 26 to catalysis, 80 to metabolism, 19 to transport, 8 to disease and 10 to reproduction (such as egg chorion and eggshell formation), etc. (Figure 6a, Appendix A). T2 down-DEGs were significantly enriched to 172 terms/pathways, with 14 relating to stress response and defense, 17 to immunity, 65 to metabolism, 8 to genetic information, 5 to disease, etc. (Figure 6b, Appendix A). T3 down-DEGs were significantly enriched to 196 terms/pathways, with 13 relating to stress response and defense, 23 to immunity, 24 to catalysis, 76 to metabolism, 7 to genetic information, 6 to signaling transduction, etc. (Figure 6c, Appendix A). 

On specific gene level, a large number DEGs relating to oxidative stress (125 up, 302 down), reproduction (27 up, 105 down), ion channels (32 up, 68 down), neuropeptide (21 up, 73 down) and P450 (16 up, 53 down) were found in the three blue light treatments, with most of them being downregulated (Figure 6d–f). Quite a number of DEGs relating to transcription factors (162 up, 55 down), protein kinase (39 up, 31 down), DNA repair (27 up, 16 down) and circadian clock (24 up, 22 down) were also found in the three blue light treatments, with most of them being upregulated.

#### 3.3.6. Validation of RNAseq Results via qPCR

A total of 29 DEGs were randomly selected from different comparison groups and their expression levels were verified using qPCR. The results indicated that the transcript levels of these DEGs (Figure 7) were similar to the RNAseq results, indicating that the RNAseq results were reliable.

## 4. Discussion

In the present study, we found that blue light exposure negatively affected the development, survival and reproductive fitness of *S. frugiperda*, with longer exposure resulting in harmful effects (Figure 1 and Figure 2). Previous studies have shown that blue light irradiation can kill insects and that the lethal wavelengths of blue light are species-specific [6,7,16]. A few studies also demonstrated that blue light exposure negatively affected the reproduction of insects [53,58,59]. However, the mechanisms of blue light toxicity in insects remain poorly understood. In *D*. *melanogaster*, blue light exposure causes brain neurodegeneration and impairments in mitochondrial respiratory function, which may directly or indirectly affect the physiology and survival of flies [24,25]. A study in *Drosophila* also found that the senolytic drug quercetin (an anti-aging medicine that can remove senescent cells) can significantly extend the survival of flies and increase the egg production of females under blue light irradiation, suggesting that blue light irradiation may accelerate the aging of insects [58].

A recent study in *D. citri* based on transcriptomic analysis further revealed that blue light stress caused remarkable transcriptional changes, where the blue light (400 nm) stress caused 1773 DEGs (841 up and 932 down) in the head [37]. In the present study, blue light (465–475 nm) stress in *S. frugiperda* females induced fewer DEGs in the head, but more DEGs in the thorax and many more DEGs in the abdomen (Figure 3). Moreover, the number and types of DEGs were irradiation-duration- and tissue-specific, with moderate-duration irradiation (T2) causing the largest number of DEGs in head, whereas long-duration irradiation (T3) caused the largest number of DEGs in the thorax, while short-duration irradiation (T1) caused the largest number of DEGs in abdomen. 

Functional enrichment analysis revealed that, in the head, T1 and T3 DEGs only enriched a few pathways (Figure 4a,c), whereas T2 DEGs enriched many more (50 enriched in up-DEGs and 21 in down-DEGs) terms/pathways, with most of them relating to metabolism, cellular process and disease (Figure 4b). In *D. citri*, blue-light-stress-induced DEGs in the head enriched to 76 terms/pathways, with quite a number of them relating to protein denaturation, oxidative stress and disease [37]. In the present study, we further tested the transcriptional response to blue light irradiation in the thorax and abdomen of *S. frugiperda* females. In the thorax, relatively more terms/pathways were enriched to stress detection and response, cellular process, metabolism, catalysis and disease, with more of them being enriched in up-DEGs in T1 treatment and more in down-DEGs in T3 treatment. In the abdomen, all blue light treatments enriched a large number (190–292) of terms/pathways, with most of them enriched in metabolism, catalysis and cellular process, and most of them (81–91%) enriched in down-DEGs.

Although the number of oxidative-stress-related terms/pathways is not prominent, the number of oxidative-stress-related DEGs is predominant in different tissues under different blue light treatments, with most of them being upregulated within the head (Figure 4d–f) and T1 and T2 thorax (Figure 5d,e), whereas most of them are downregulated within the T3 thorax (Figure 5f) and abdomen (Figure 6d–f). Oxidative stress is recognized as the disruption of the balance between antioxidant defenses and the synthesis of ROS [60]. Abiotic stress, such as UV radiation, extreme temperatures, or chemicals, can promote the synthesis and accumulation of ROS. ROS are not harmful at very low levels and play roles in signal transduction and stress defense by increasing oxidant production and cell antioxidant capacity [61,62]. However, they are harmful to DNA and proteins at higher levels [63].

In transcription-factor-related DEGs (Appendix A), quite a number of upregulated zinc finger proteins (ZNFs) were found in blue-light-stressed females. ZNFs interact with DNA, RNA and other proteins, and are involved in the regulation of transcription and cellular processes, such as ubiquitin-mediated protein degradation, DNA repair and cell migration [64]. Many ZNFs have been proven to be associated with biotic and abiotic stresses [65,66]. In *Apis cerana*, oxidative stress increased the expression of ZFP41 and the overexpression of this gene in vivo increased tolerance to oxidative stress [67]. This evidence provides a theoretical basis for further studying the function of ZNFs in stress resistance.

The insect abdomen is the reproductive center and food digestion site, as well as the energy and nutrient reservoir in the form of fat bodies [36]. Basic metabolic processes account for about half of an individual’s energy expenditure and are associated with individual fitness and survival [68]. As mentioned above, abiotic stress may prompt insects to use antioxidant defenses, such as superoxide dismutase, ascorbic acid and glutathione-S-transferase, to diminish cellular damage [69]. Therefore, reducing of metabolic rate and other biological processes in the abdomen may be a trade-off between anti-stress defense and another process or strategy to escape stressful conditions [70]. In addition, blue light stress reduced the bodyweight and reproduction output of females (Figure 1 and Figure 2), which may accordingly be accompanied by the occurrence of lower nutrient reserve and a lower metabolic rate.

A great number (132) of reproduction-related DEGs were also shown in the abdomen of females, with most (76%) of them being downregulated (Appendix A). These down-DEGs (including a large number of chorion proteins) were mainly related to egg development, which may be partially linked to reduced fecundity in blue-light-stressed females (Figure 2). In insects, the chorion plays a basic role in protecting embryos from external factors during development and allowing gas exchange for respiration [71]. 

Abiotic stress generally causes the upregulation of HSPs in insects [72,73]. HSPs are molecular chaperones, which also help the organisms against a variety of stresses by coping with the stress-induced denaturation of proteins [74]. In *D. citri*, a number of Hsp70s and one *alpha-crystallin* (also a chaperone) were found to be upregulated in blue-light-stressed groups [37]. In the present study, 37 HSP DEGs were found, but only a few (8) of them were upregulated in blue-light-stressed groups. This may also be due to the above mentioned trade-offs or other mechanisms that are still unknown.

A total of 53 human disease pathways were enriched in this study, but most of them (79%) were enriched in down-DEGs. A further analysis detected that 15 of these pathways were infectious diseases (bacterial, viral or parasitic) and most of them (11/15) were enriched in down-DEGs. The antibacterial properties of blue light have been studied for a long time and recent extensive studies have shown that blue light can be a broad-spectrum antimicrobial [75,76,77]. Thus, blue light exposure may reduce infectious diseases in *S. frugiperda*.

Moreover, the present study in *S. frugiperda* and studies in other insect species have partially revealed the effect process and mechanism of blue light stress, in which the photoreceptors and other pathways (such as ion channels) function in detecting and responding to blue light stress. This then causes the action and even the degeneration of brain neurons [25] and other signal transduction pathways (such as the MAPK signaling pathway) and molecular processes (such as oxidative stress, DNA repair and circadian clock) and eventually causes molecular and physiological changes in the whole body. Apart from the eyes, insects have photosensitive organs in various parts of their bodies, such as socketed hairs (sensory setae) in lepidopterans and other insects that are distributed across most parts of the body, which may be sensitive to sounds, touch and light [78]. This may be a reason for the upregulation of light stress detection and response-related genes and pathways in the thorax (Figure 5a). Protein subsets of ligand- and voltage-gated (or -dependent) ion channels, such as calcium, sodium and transient receptor potential (TRP), also function in detecting mechanical and chemical insults and generating neuronal action potentials [79,80,81]. Protein-kinase-based cellular signal transduction, especially the MAPK (mitogen activated protein kinase) pathway, may be activated to regulate the stress response, including DNA repair, the antioxidant response and cell cycle regulation [82,83].

Importantly, the above studies clearly indicate that blue light irradiation can have negative impacts on insects and that strong blue light irradiation is lethal to insects. These results suggest that blue light can be used to develop environmentally friendly control strategies. For example, blue light has been applied to repel the whitefly *T. vaporariorum* from its host [41]. However, further research is still needed to promote the application of blue light irradiation in pest control, such as by combining blue light irradiation with insecticide control or pheromone-based mating disruption. Moreover, blue light can be considered for the control of insect pests and plant pathogens simultaneously. 

In conclusion, blue light irradiation significantly reduced the survival and reproductive success of *S. frugiperda* and showed dose-dependent effects. Transcriptomic analysis further found that blue light irradiation induced dose- and tissue-specific transcriptional responses. These results also suggest that the irradiation-induced reduction in survival and reproduction may be due to direct damage to the cell and molecular function, as well as indirect effects from increased expenditure on stress-resistant progress. These results provide novel evidence on the irradiation stresses and responses of insects and contribute to the development of environmentally friendly control strategies for *S. frugiperda*.

## Figures and Tables

**Figure 1 insects-15-00010-f001:**
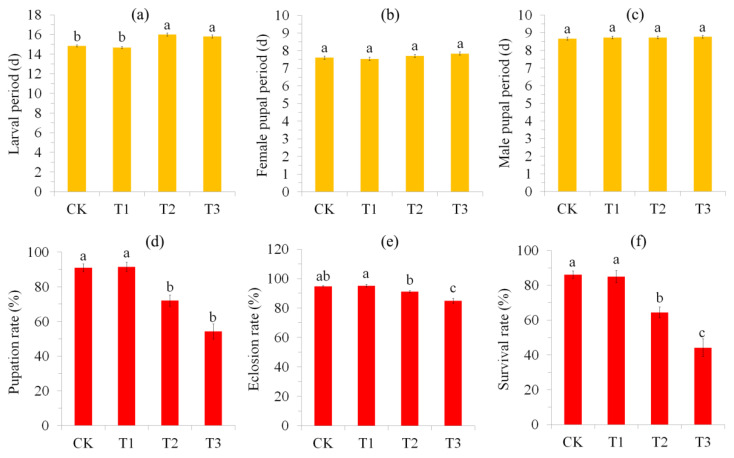
Effect of blue light exposure on the developmental periods and survival of *S. frugiperda*. (**a**–**c**) are larval period, female pupal period and male pupal period, respectively; (**d**–**f**) are pupation rate, eclosion rate and survival rate, respectively. Error bars are standard error (SE). In each subgraph, bars with different letters are significantly different (*p* < 0.05; tested using Tukey’s studentized range test).

**Figure 2 insects-15-00010-f002:**
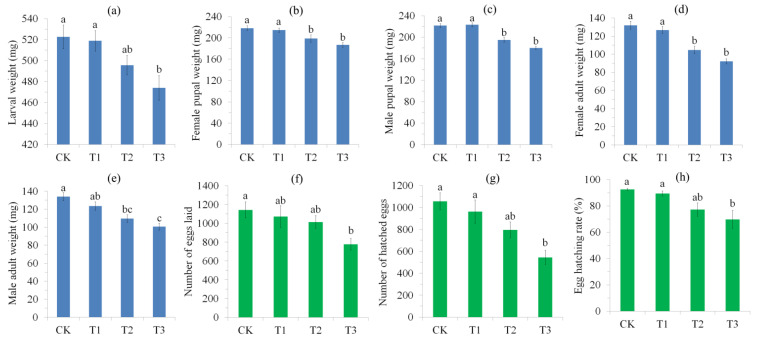
Effect of blue light exposure on the bodyweight and reproduction of *S. frugiperda*. (**a**–**e**) are larval bodyweight, female pupal bodyweight, male pupal bodyweight, female adult bodyweight and male adult bodyweight, respectively; (**f**–**h**) are number of eggs laid, number of hatched eggs and egg hatching rate, respectively. Error bars are standard error (SE). In each subgraph, bars with different letters are significantly different (*p* < 0.05; tested by Tukey’s studentized range test).

**Figure 3 insects-15-00010-f003:**
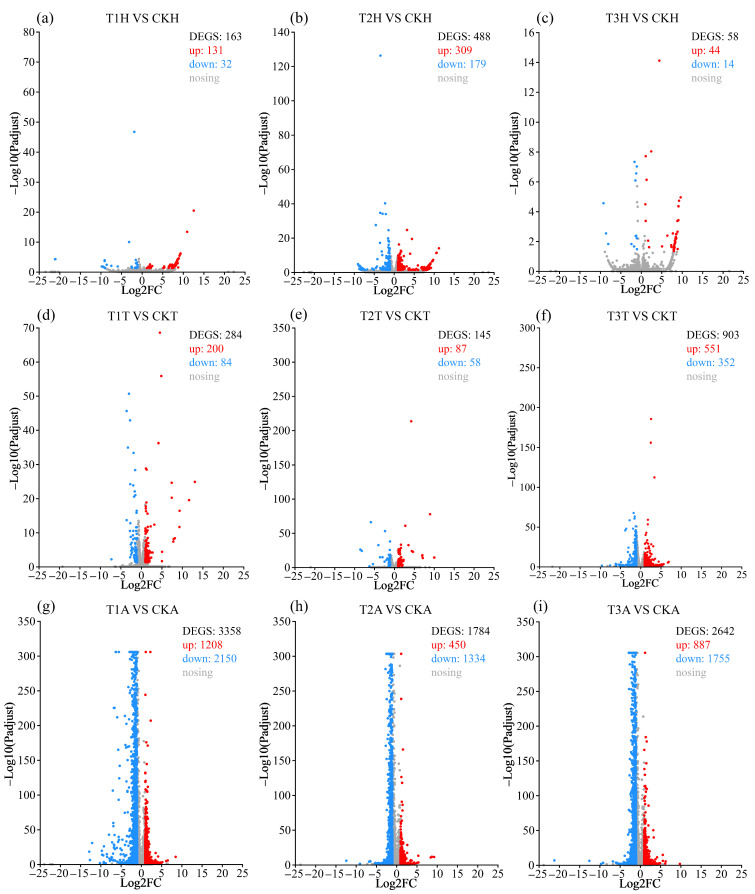
Volcanic plots of DEGs induced by blue light stress in *S. frugiperda*. (**a**–**c**) are transcription changes in female head caused by T1, T2 and T3 treatments, respectively; (**d**–**f**) are transcription changes in female thorax caused by T1, T2 and T3 treatments, respectively; and (**g**–**i**) are transcription changes in female abdomen caused by T1, T2 and T3 treatments, respectively. Genes significantly upregulated in expression are designated by blue dots and genes significantly downregulated in expression by red dots. Genes with no significant differences in expression are indicated by gray dots. T1H, T2H and T3H refer to female head from T1, T2 and T3 treatments, respectively; T1T, T2T and T3T refer to female thorax from T1, T2 and T3 treatments, respectively; T1A, T2A and T3A refer to female abdomen from T1, T2 and T3 treatments, respectively; CKH, CKT and CKA refer to control female head, thorax and abdomen, respectively; VS, versus.

**Figure 7 insects-15-00010-f007:**
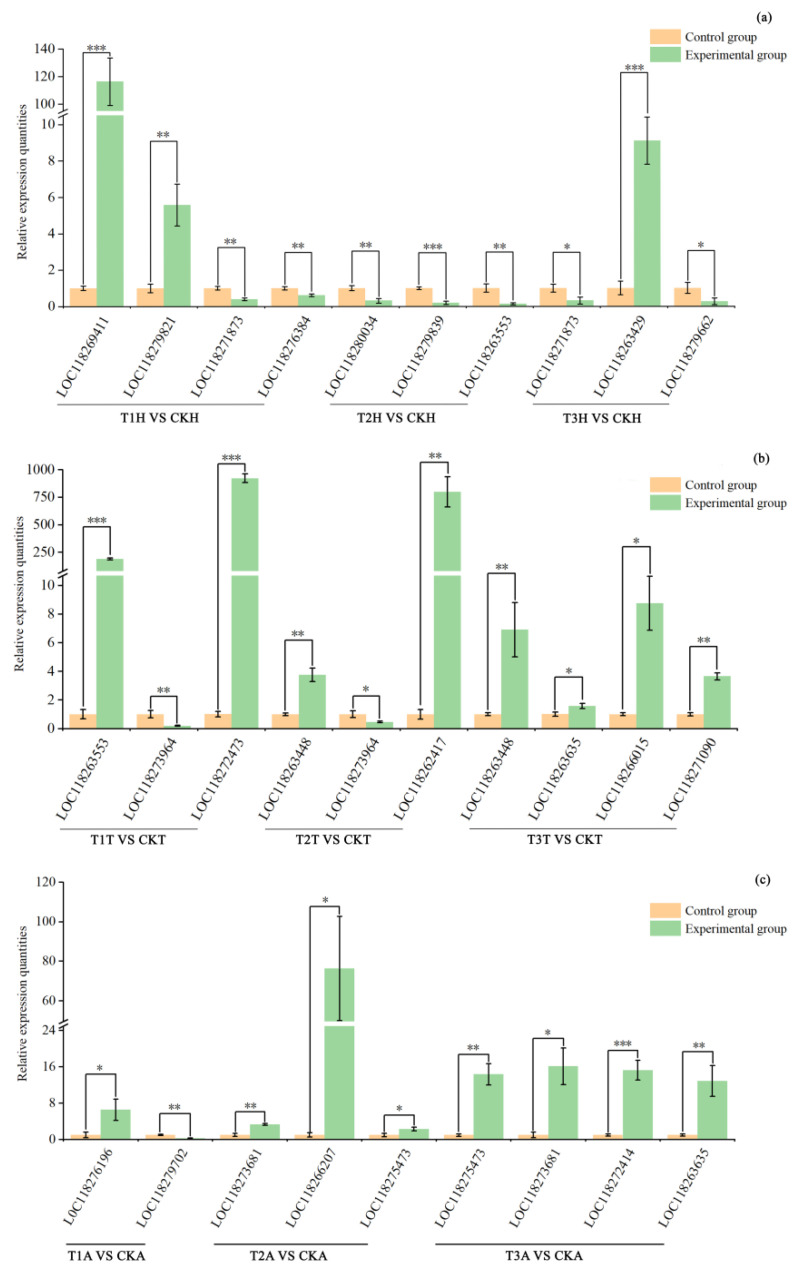
Validation of RNAseq results by qPCR. (**a**–**c**) are relative expression levels of target genes in female heads, thoraxes and abdomens, respectively. * indicates *p* < 0.05; ** indicates *p* < 0.01, *** indicates *p* < 0.001 (tested by ANOVA). T1H, T2H and T3H refer to female head from T1, T2 and T3 treatments, respectively; T1T, T2T and T3T refer to female thorax from T1, T2 and T3 treatments, respectively; T1A, T2A and T3A refer to female abdomen from T1, T2 and T3 treatments, respectively; CKH, CKT and CKA refer to control female head, thorax and abdomen, respectively; VS, versus. The codes under the X-axis are gene IDs and their annotations can be found in Appendix A.

## Data Availability

The transcriptome raw reads have been deposited into the NCBI SRA database; the accession number is PRJNA1027141. Other data generated or analyzed during this study are included in this article and its Appendix A.

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
