# Peer review of "Blue Light Exposure Caused Large-Scale Transcriptional Changes in the Abdomen and Reduced the Reproductive Fitness of the Fall Armyworm Spodoptera frugiperda"

_insects, 2023, doi:10.3390/insects15010010_

Round 1

Reviewer 1 Report

Comments and Suggestions for Authors

Liu et al. report the results of a study on the effects of exposure to blue visible light on the development and transcription profiles of Spodoptera frugiperda (Sf).
The study has serious flaws and I cannot recommend that it be published in its present form.
Major points.
(I) The authors performed three replicates of the developmental study based on 4 groups of insects (3 light treatments + control). However, they then divided a single replicate into 30 individual insects and analyzed each individual as an independent observation, which is obviously a case of pseudoreplication. This resulted in 120 pseudoreplicated observations in the ANOVA analyses rather than 12 (4 treatments x 3 replicates) which would have been the correct way to analyze these results. The authors need to calculate the mean response observed in each replicate and reanalyze the mean values (i.e. total of 12 data points from each experiment shown in Fig 1 and Fig 2.).
(II) The whole experiment depends of exposure to a certain wavelength of light. However, the authors have not measured or characterized the light spectra of the LED sources that they used and have not considered the effects of the insect cages (and possibly the glass or plastic rearing posts) that they used to contain the insects. Therefore, we have no idea of the true wavelengths of light (blue or white) that the experimental insects were exposed to.
(III) The English text has many errors and is difficult to understand.
I have written numbered points and suggestions on a scanned copy of the manuscript.
Numbered points.
1. Summary: mechanism of what?
2. Shorter blue light? Meaning shorter duration of exposure or shorter wavelength of light?
Shorter than what?
3. This is a highly detailed text that should be written in more general terms to provide the reader with an overview of general effects of blue light on biological systems.
4. Mention that Sf spread across Africa and southern Asia before reaching China and is now invading Oceania.
5. What is the importance of studying blue light in Sf? Why did you perform this study? Does this have any pest control applications? or use in mass-rearing Sf for SIT based pest control?
6a. You need to characterize the light spectra that you used. The concept of "white light" is not defined at all. The concept of "blue light" is given in relation to the manufacturer's specification of an LED bulb, but in my experience what the manufacturer says and what the bulb really emits are quite different things. You need to use a spectroradiometer to measure these spectra.
6b. You also need to measure the spectra within the insect cages to determine how cage structure (acrylic, nylon mesh, etc.) affect the transmitted spectra.
Were the insects placed in plastic containers or pots with diet?
How did the plastic affect the transmitted light spectrum?
If insects were not reared individually in pots, there was probably a significant degree of cannibalism which is itself likely to be stressful to the insects.
7. Please use the word "control" in the text although you can use "CK" in figures if you explain its use.
8. You clearly state that you performed THREE replicates of 120 insects. You cannot then generate 30 - 60 new replicates (individual insects) from each single replicate as these, by definition, are pseudoreplicates. You need to AVERAGE the responses of all insects within each replicate and then perform your analysis using the three mean values from the three replicates.
In the future I would suggest using fewer insects and increasing the number of replicates to account for batch-to-batch variation in insect responses, as this can have a strong effect on observed variation. For example, an experiment involving 4 treatment groups (three treatments + a control) should usually be replicated 6 - 10 times to provide adequate power for the statistical analyses.
9. Did you test normality or equality of variances prior to ANOVA? How was this achieved?
10. Why were percentages arcsine transformed? The pupation, eclosion and survival percentages in Fig 1 look normally distributed to me? Did you have an issue with heteroscedasticity?
11. The F statistics (ANOVAs) are obviously erroneous because you had only 3 replicates, so a residual(error) degrees of freedom value of 116 indicates that you have a pseudoreplication issue. These data need to be reanalyzed. Section 3.1, also numbered 3.1 on line 204 and 221, i.e. data in Fig 1 and Fig 2 need reanalysis.
12. Please explain your use of codes in the Figure legend. Fig 3, Fig 4, Fig 5, Fig 6, Fig 7.
13. Please indicate which test was applied here to generate P values.
14. Are you describing experiments with MONOCHROMATIC light? Or was this the peak wavelength used in the experiment?
15. You are correct that blue light can be antimicrobial. There are several rather old studies (1960s - 1990s) that demonstrate this effect in baculoviruses, Bacillus thuringiensis and entomopathogenic fungi that you should cite here, as all these types of pathogens infect and kill Sf.
16. Are there any practical applications of your findings to pest control targeted at Sf?
17. The supplemental material includes a lot of abbreviations that are not explained in the supplemental material.  Also, some of the supplemental images are unnecessarily large image files that should be reduced to KB size rather than MB size.

Comments on the Quality of English Language

Needs editing for grammar and clarity.

Author Response

Comments and Suggestions for Authors:

Liu et al. report the results of a study on the effects of exposure to blue visible light on the development and transcription profiles of Spodoptera frugiperda (Sf).
The study has serious flaws and I cannot recommend that it be published in its present form.

Our answer: We thank you for the constructive comments and correction suggestions to our MS. We have now revised the paper very carefully according to them.

Major points.
(I) The authors performed three replicates of the developmental study based on 4 groups of insects (3 light treatments + control). However, they then divided a single replicate into 30 individual insects and analyzed each individual as an independent observation, which is obviously a case of pseudoreplication. This resulted in 120 pseudoreplicated observations in the ANOVA analyses rather than 12 (4 treatments x 3 replicates) which would have been the correct way to analyze these results. The authors need to calculate the mean response observed in each replicate and reanalyze the mean values (i.e. total of 12 data points from each experiment shown in Fig 1 and Fig 2.).

Our answer: S. frugiperda larvae have cannibalism behavior, and thus were reared in cells individually. We initially set three replicates for each treatment to test the percentages of pupation, eclosion and survival, with one replicate using 120 larvae. We used a large number of (120) larvae in each replication to ensure sufficient survival of pupae and adults for subsequent eclosion and reproduction testing. Our preliminary research and previous reports suggested that blue light may lead to high mortality in insects. To save time, we then randomly selected 30 larvae, 30 male pupae and 30 female pupae, and 30 male adult and 30 female adult from each of the above treatments at the same time during above survival tests for developmental period and bodyweight tests for each of the life-stages. We also randomly selected 20 pairs of adults from each of the treatments for fecundity tests. Therefore, the df is 3,8 for pupation rate, eclosion rate and survival rate, and 3,116 for developmental period and bodyweight tests, and 3,76 for fecundity tests. We have now provided detailed information for this in M&M.

(II) The whole experiment depends of exposure to a certain wavelength of light. However, the authors have not measured or characterized the light spectra of the LED sources that they used and have not considered the effects of the insect cages (and possibly the glass or plastic rearing posts) that they used to contain the insects. Therefore, we have no idea of the true wavelengths of light (blue or white) that the experimental insects were exposed to.

Our answer: We agree and have now provided detailed information for this in the M&M.

(III) The English text has many errors and is difficult to understand.
I have written numbered points and suggestions on a scanned copy of the manuscript.
Numbered points.
1. Summary: mechanism of what?

Our answer: Thanks and revised.

  1. Shorter blue light? Meaning shorter duration of exposure or shorter wavelength of light?
    Shorter than what?

Our answer: It means shorter duration of exposure. Revised.

  1. This is a highly detailed text that should be written in more general terms to provide the reader with an overview of general effects of blue light on biological systems.

Our answer: Agree and revised.

  1. Mention that Sf spread across Africa and southern Asia before reaching China and is now invading Oceania.

Our answer: Agree and done.

  1. What is the importance of studying blue light in Sf? Why did you perform this study? Does this have any pest control applications? or use in mass-rearing Sf for SIT based pest control?

Our answer: Agree and provided more information on this in Introduction.

6a. You need to characterize the light spectra that you used. The concept of "white light" is not defined at all. The concept of "blue light" is given in relation to the manufacturer's specification of an LED bulb, but in my experience what the manufacturer says and what the bulb really emits are quite different things. You need to use a spectroradiometer to measure these spectra.

Our answer: Agree and have now provided detailed information here.

6b. You also need to measure the spectra within the insect cages to determine how cage structure (acrylic, nylon mesh, etc.) affect the transmitted spectra.
Were the insects placed in plastic containers or pots with diet?
How did the plastic affect the transmitted light spectrum?
If insects were not reared individually in pots, there was probably a significant degree of cannibalism which is itself likely to be stressful to the insects.

Our answer: Agree and have now provided detailed information here.

  1. Please use the word "control" in the text although you can use "CK" in figures if you explain its use.

Our answer: Agree and done

  1. You clearly state that you performed THREE replicates of 120 insects. You cannot then generate 30 - 60 new replicates (individual insects) from each single replicate as these, by definition, are pseudoreplicates. You need to AVERAGE the responses of all insects within each replicate and then perform your analysis using the three mean values from the three replicates.
    In the future I would suggest using fewer insects and increasing the number of replicates to account for batch-to-batch variation in insect responses, as this can have a strong effect on observed variation. For example, an experiment involving 4 treatment groups (three treatments + a control) should usually be replicated 6 - 10 times to provide adequate power for the statistical analyses.

Our answer: Please see our response as above. More detailed information was provided here now.

  1. Did you test normality or equality of variances prior to ANOVA? How was this achieved?

Our answer: These have been done before ANOVA, and more information on this has been provided now.

  1. Why were percentages arcsine transformed? The pupation, eclosion and survival percentages in Fig 1 look normally distributed to me? Did you have an issue with heteroscedasticity?

Our answer: The data of some treatments were not normally distributed and thus were transformed to normalize the data. The arcsine square root-transformation is usually used to normalized the percentage data. More information on this has been provided here now.

  1. The F statistics (ANOVAs) are obviously erroneous because you had only 3 replicates, so a residual(error) degrees of freedom value of 116 indicates that you have a pseudoreplication issue. These data need to be reanalyzed. Section 3.1, also numbered 3.1 on line 204 and 221, i.e. data in Fig 1 and Fig 2 need reanalysis.

Our answer: Please see our above response for experimental design and statistics.

  1. Please explain your use of codes in the Figure legend. Fig 3, Fig 4, Fig 5, Fig 6, Fig 7.

Our answer: Thanks and done.

  1. Please indicate which test was applied here to generate P values.

Our answer: Done.

  1. Are you describing experiments with MONOCHROMATIC light? Or was this the peak wavelength used in the experiment?

Our answer: Thanks and provided relevent information.

  1. You are correct that blue light can be antimicrobial. There are several rather old studies (1960s - 1990s) that demonstrate this effect in baculoviruses, Bacillus thuringiensis and entomopathogenic fungi that you should cite here, as all these types of pathogens infect and kill Sf.

Our answer: We thank this constructive suggestion and have now provided relevant information and references here now.

  1. Are there any practical applications of your findings to pest control targeted at Sf?

Our answer: Agree and have now provided more information on this in the last two paragraphs in the Discussion.

  1. The supplemental material includes a lot of abbreviations that are not explained in the supplemental material.  Also, some of the supplemental images are unnecessarily large image files that should be reduced to KB size rather than MB size.

Our answer: Thanks and have now provided full names for all abbreviations, and also provided small image files.

Comments on the Quality of English Language

Needs editing for grammar and clarity.

Our answer: We thank you again for the comments and corrections. We have now revised the paper very carefully according to them and comments from other reviewers, which have substantially improved the quality of this paper.

Reviewer 2 Report

Comments and Suggestions for Authors

The manuscript discusses the species-specific response to blue light exposure in S. frugiperda and refers to previous studies highlighting differential effects on various insect species. Investigating and understanding the underlying genetic and physiological factors that contribute to these species-specific responses is crucial for generalizing the findings and predicting the impact of blue light exposure on diverse insect populations. Overall, manuscript is well written and quite clear. Meanwhile, I have some concerns and question for authors. 

Major concerns:

Firstly, the manuscript describes the transcriptional changes induced by blue light stress in different tissues of S. frugiperda females. I am concerned here not to focus on the variability in the number and types of differentially expressed genes (DEGs) across tissues and exposure durations and simply presenting DEGs for individual tissues. Exploring the reasons behind tissue-specific responses and understanding the functional implications of these transcriptional changes could enhance the interpretation of the study's results. Secondly, given the potential negative effects of blue light exposure on the reproduction and survival of S. frugiperda, a major scientific concern could focus on the practical implications for pest management strategies. Understanding how manipulating light conditions could be employed as an environmentally friendly pest control method or integrated into existing pest management practices would be of significant interest to agricultural and ecological communities. Thus, authors are expected to provide a simple paragraph detailing the future road map for others about how to read this data in more ecological framework, e.g., pest management, blue light disease management etc. 

Minor comments:

Most of the minor comments are marked on the pdf file. In brief, major focus needs to be on the statistical method representation and conclusion. 

Author Response

Comments and Suggestions for Authors:

The manuscript discusses the species-specific response to blue light exposure in S. frugiperda and refers to previous studies highlighting differential effects on various insect species. Investigating and understanding the underlying genetic and physiological factors that contribute to these species-specific responses is crucial for generalizing the findings and predicting the impact of blue light exposure on diverse insect populations. Overall, manuscript is well written and quite clear. Meanwhile, I have some concerns and question for authors. 

Major concerns:

Firstly, the manuscript describes the transcriptional changes induced by blue light stress in different tissues of S. frugiperda females. I am concerned here not to focus on the variability in the number and types of differentially expressed genes (DEGs) across tissues and exposure durations and simply presenting DEGs for individual tissues. Exploring the reasons behind tissue-specific responses and understanding the functional implications of these transcriptional changes could enhance the interpretation of the study's results. Secondly, given the potential negative effects of blue light exposure on the reproduction and survival of S. frugiperda, a major scientific concern could focus on the practical implications for pest management strategies. Understanding how manipulating light conditions could be employed as an environmentally friendly pest control method or integrated into existing pest management practices would be of significant interest to agricultural and ecological communities. Thus, authors are expected to provide a simple paragraph detailing the future road map for others about how to read this data in more ecological framework, e.g., pest management, blue light disease management etc. 

Our answer: We thank you for the constructive comments and correction suggestions to our MS. We have now provided more information and discussion on the reasons behind blue light stress responses and its potential application in pest control based on our data in both the Introduction and Discussion parts.

Minor comments:

Most of the minor comments are marked on the pdf file. In brief, major focus needs to be on the statistical method representation and conclusion. 

Our answer: Detailed information for experimental design and statistics were provided in M&M, and a conclusion was added in the end of Discussion. We also revised the paper very carefully according to all other comments, which have substantially improved the quality of this paper.

We thank you again for the comments and corrections.

Round 2

Reviewer 1 Report

Comments and Suggestions for Authors

The authors have addressed my concerns and have improved their manuscript.

My only suggestion is to ask them to indicate how they tested for normality and homoscedasticity of their data (line 220 - 221).

Comments on the Quality of English Language

Requires editing.

Author Response

The authors have addressed my concerns and have improved their manuscript.

Our answer:We thank you very much for your comments to our MS again.

My only suggestion is to ask them to indicate how they tested for normality and homoscedasticity of their data (line 220 - 221).

Our answer:We agree and have now added this information in the MS.

Reviewer 2 Report

Comments and Suggestions for Authors

The manuscript has undergone significant improvement during the revision process, resulting in an enhanced overall presentation. However, it has come to my attention that the quality of the figures appears to be distorted for reasons yet to be determined. In light of this, I strongly encourage the authors to upload high-quality images to ensure optimal clarity and fidelity in the visual elements accompanying the manuscript. This will contribute to maintaining the professional standard and visual integrity of the publication. 

Comments on the Quality of English Language

Marked on the PDF file! 

Author Response

The manuscript has undergone significant improvement during the revision process, resulting in an enhanced overall presentation. However, it has come to my attention that the quality of the figures appears to be distorted for reasons yet to be determined. In light of this, I strongly encourage the authors to upload high-quality images to ensure optimal clarity and fidelity in the visual elements accompanying the manuscript. This will contribute to maintaining the professional standard and visual integrity of the publication. 

Our answer:We are very grateful for your reviewing and comments to our MS again. We have now revised the paper according to the comments in the PDF file. We have now changed the figures in the MS to high resolution ones and uploaded high resolution supplemental figures.